# Distinct Mechanisms of Endomembrane Reorganization Determine Dissimilar Transport Pathways in Plant RNA Viruses

**DOI:** 10.3390/plants11182403

**Published:** 2022-09-15

**Authors:** Andrey G. Solovyev, Anastasia K. Atabekova, Alexander A. Lezzhov, Anna D. Solovieva, Denis A. Chergintsev, Sergey Y. Morozov

**Affiliations:** 1A. N. Belozersky Institute of Physico-Chemical Biology, Moscow State University, 119992 Moscow, Russia; 2Department of Virology, Biological Faculty, Moscow State University, 119234 Moscow, Russia; 3All-Russia Research Institute of Agricultural Biotechnology, 127550 Moscow, Russia

**Keywords:** plant virus, virus movement, movement protein, plasmodesmata, viral replication complexes, endomembrane system, endosome

## Abstract

Plant viruses exploit the endomembrane system of infected cells for their replication and cell-to-cell transport. The replication of viral RNA genomes occurs in the cytoplasm in association with reorganized endomembrane compartments induced by virus-encoded proteins and is coupled with the virus intercellular transport via plasmodesmata that connect neighboring cells in plant tissues. The transport of virus genomes to and through plasmodesmata requires virus-encoded movement proteins (MPs). Distantly related plant viruses encode different MP sets, or virus transport systems, which vary in the number of MPs and their properties, suggesting their functional differences. Here, we discuss two distinct virus transport pathways based on either the modification of the endoplasmic reticulum tubules or the formation of motile vesicles detached from the endoplasmic reticulum and targeted to endosomes. The viruses with the movement proteins encoded by the triple gene block exemplify the first, and the potyviral system is the example of the second type. These transport systems use unrelated mechanisms of endomembrane reorganization. We emphasize that the mode of virus interaction with cell endomembranes determines the mechanism of plant virus cell-to-cell transport.

## 1. Introduction

The plant endomembrane system (EMS) is essential for biosynthetic processes and targeted distribution of macromolecules and other compounds in cells. The major compartments comprising the plant EMS include the nuclear envelope, the endoplasmic reticulum (ER), the Golgi apparatus, the *trans*-Golgi network (TGN), the endosomes, and the vacuole [1]. Distinct membrane compartments are connected to each other through vesicular trafficking, the process in which membrane vesicles serving as transport containers are detached from source EMS compartments and directionally delivered to target membrane compartments [1,2]. In particular, the trafficking pathways between the ER and Golgi are mediated by COPII- and COPI-coated vesicles in the case of ER-to-Golgi and retrograde Golgi-to-ER transport, respectively [3,4,5], whereas the transport between the TGN, the vacuole, the endosomes, and the plasma membrane (PM) is mediated by clathrin-coated vesicles [2,6]. The vesicular transport serves for the intracellular delivery of proteins contained within the EMS compartments (e.g., in the ER lumen, Golgi cisternae, or endosomes), as well as membrane components and membrane-embedded proteins [1]. Additionally, distinct membrane compartments are interconnected by membrane contact sites (MCSs), the sites of interactions of the ER membrane with endosomes, peroxisomes, Golgi, vacuoles, the PM, as well as organelles such as mitochondria and plastids [7]. The MCSs are proposed to constitute a non-vesicular inter-organellar communication pathway [8] required for signal transmission and exchange of lipids and possibly proteins between the ER and other organelles [9,10,11].

Plant viruses exploit the EMS for their replication and cell-to-cell transport [12]. The replication of viral (+)RNA-genomes occurs in the cytoplasm, and the virus progeny can be transported from infected cells to surrounding healthy cells through plasmodesmata (PD), plant-specific intercellular channels that traverse cell walls and provide direct contacts of the cytoplasm and the EMS of neighboring cells [13,14,15]. The PD include the PM lining the PD channel, and the desmotubule, a constricted ER tubule continuous with the ER of two adjoining cells, spanning the pore on its axis [14]. In the PD channel, the PM and the desmotubule are tethered together by protein elements characteristic of MCSs [15,16,17]. These protein tethers are believed to contribute to regulating the size of the cytoplasmic sleeve between the PM and the desmotubule that determines the PD permeability, or the PD size exclusion limit (SEL) [18]. However, the major mechanism of the regulation of the PD SEL involves the local callose deposition in the cell wall regions surrounding the PD that results in the PD ‘closure’ and degradation of PD-associated callose by β-1,3-glucanase, leading to the PD SEL increase [19]

The cell-to-cell transport of plant virus genomes requires virus-encoded movement proteins (MPs) essential for the targeting of progeny viral genomic RNA or virions to PD and their transport through the PD channels to neighboring cells [20]. The *Tobacco mosaic virus* (TMV) MP is the paradigm MP, the first-discovered and currently best-studied among proteins of this function [12]. Two basic properties of the TMV MP were described soon after its discovery, namely the abilities (i) to bind RNA and (ii) to accumulate in PD, increase the PD SEL, and move cell to cell [21]. Accordingly, the TMV MP has been hypothesized to bind viral RNA and form transport-competent viral ribonucleoprotein complexes (vRNPs) capable of trafficking through the PD channel dilated by the TMV MP [21]. Numerous further studies have demonstrated a tight link between the TMV replication and transport, as well as a close association of both these processes with the ER and the cytoskeleton. In particular, the TMV MP is shown to localize to the ER structures, to interact with microtubules, and to co-localize with viral RNA [12,22,23].

According to current models of TMV cell-to-cell transport, at the initial infection stages, the viral replication complexes (VRCs), which contain viral RNA, replicase, and MP, are formed on the ER membranes in association with microtubules [12,24]. These early VRCs are mobile and move in the cytoplasm along the ER in an actin-dependent manner, reaching the PD proximity [25,26,27]. The TMV MP is believed to be the protein that targets VRCs to PD, as the MP on its own is capable of PD targeting, most probably due to a specific PD localization signal that is mapped to the protein N-terminal region. This MP region is involved in the MP interaction with SYTA, a transmembrane protein of the synaptotagmin family located in the MCSs, and is involved in virus cell-to-cell transport, suggesting that SYTA may play a role in the VRC targeting to PD [28,29,30,31,32]. Further, upon replication in the PD-targeted VRCs, the virus RNA progeny is likely directed to transport through the PD channels [33]. 

The TMV MP has been initially predicted to be an integral membrane protein [34,35]. However, according to further experimental analysis of the MP topology in the ER membrane, the protein peripherally associates with the cytoplasmic face of the ER [36]. Additionally, the TMV RNA injected in plant cells forms granules anchored to the ER network, and the methylguanosine cap at the 5′-terminus of virus RNA is required for the anchoring [33]. Therefore, as one of possible models, the VRC formation may be initiated by virus RNA anchoring at the ER membrane and further continued upon local translation of viral membrane-interacting proteins, namely the TMV MP and the 126K protein, a TMV replicase component known to associate with the ER and co-translationally bind the 5′-terminal region of the TMV genomic RNA [37]. However, these and other available observations clarify neither the TMV VRC structure nor the mechanism of the VRC formation of the ER membranes of TMV-infected cells. 

In this review, focused mainly on the intracellular transport of viral genomes to PD, we discuss two distinct TMV-unrelated plant virus transport systems, for which dissimilar mechanisms of reorganization of cell endomembranes for virus transport are currently uncovered and characterized in detail. Further, we make parallels between the mechanisms of the intracellular transport of virus-specific RNAs and cellular mRNAs. 

## 2. TGB/BMB: The ER-Based Pathway

### 2.1. Properties of TGB Proteins

In plant virus-encoded multicomponent transport systems, i.e., those involving two or more MPs, multiple functions performed by the single MP of TMV are distributed among several proteins, showing a ‘division of labor’. A well-studied example of such a multicomponent transport system is a ‘triple gene block’ (TGB) [38]. TGB is an evolutionary conserved gene module found in many (+)RNA plant viruses of families *Virgaviridae*, *Alphaflexiviridae*, *Betaflexiviridae*, and *Benyviridae*. TGB consists of three partially overlapping genes encoding MPs termed TGB1, TGB2, and TGB3, which have characteristic structural features [39].

TGB1 protein contains an NTPase/helicase domain belonging to a divergent lineage of ‘accessory’ viral SFI (superfamily I) helicases that likely have evolved after a duplication of a replicative RNA helicase domain [39,40,41]. Accordingly, TGB1 proteins are experimentally shown to have the activities of NTPase [39] and RNA helicase [42]. As demonstrated by the mutagenesis of conserved motifs of the NTPase/helicase domain, the enzymatic activities of TGB1 are essential for its function in cell-to-cell transport [43,44,45]. Additionally, TGB1 protein exhibits a non-specific RNA-binding activity, which is believed to be involved in the formation of vRNPs with virus genomic RNA [45,46,47,48]. The TGB1 protein has also been shown to act as a suppressor of RNA silencing, and this activity can be linked to the efficiency of virus transport [49,50].

Based on comparative analyses, two types of TGB are distinguished, potex-like and hordei-like TGBs, termed by the names of two well-studied virus genera *Potexvirus* and *Hordeivirus*, having distinct TGBs [39]. In hordei-like TGBs, the TGB1 protein has an additional N-terminal domain, absent in potex-like TGBs, which is involved in the systemic transport of virus infection via the phloem [46,51,52]. The fundamental difference between viruses with potex-like and hordei-like TGBs is the dependence of virus cell-to-cell transport on viral capsid protein (CP) and genome encapsidation, known to be essential for transport in the case of potex-like but not hordei-like TGB [53]. As shown for *Potato virus X* (PVX), the type potexvirus, several TGB1 molecules can bind one end of PVX filamentous virions that renders them transport competent and enables the translation of virion-derived RNAs after the virus has moved cell to cell [54]. By contrast, encapsidation is not required for the transport of *Barley stripe mosaic virus* (BSMV), the type hordeivirus, and its genomic RNA is transported through the PD as a complex with TGB1, which appears to be the sole protein component of such a complex [55,56]. Therefore, the TGB1 roles in the cell-to-cell transport are not identical in potex- and hordei-like TGBs. 

TGB2 and TGB3 are small integral membrane proteins, and mutations disrupting their association with membranes block virus transport [57,58]. TGB2 has two hydrophobic domains with a conserved region between them. As shown for the TGB2 of both potex- and hordei-like TGB, the N- and C-termini are exposed to the cytoplasm, while the central hydrophilic loop is exposed to the ER lumen [59,60]. However, as suggested by further analysis of TGB2 functional interactions, the protein central region can be located in the cytoplasm [61] (discussed below). The TGB3 protein of the potex-like TGBs has a single transmembrane domain, whereas the hordei-like TGB3 protein is sequence unrelated to that in the potex-like TGBs and has two transmembrane domains [39,62].

In subcellular localization experiments, the TGB2 and TGB3 proteins are found in the ER and/or ER-derived membrane structures. In particular, TGB2 fused to fluorescent proteins is found to be localized, depending on the experimental system, either in cortical ER tubules or numerous ER-related tiny vesicles, whereas TGB3 is found in both intracellular ER-associated structures and ER-derived membrane bodies located at the cell periphery in close vicinity of PD [63]. These structures are called hereafter ‘PD-associated membrane bodies’ (PAMBs). The TGB3 localization to PAMBs requires specific signals mapped in the TGB3 proteins of several viruses [64,65,66,67], and the TGB3 intracellular transport to PAMBs bypasses the conventional secretory pathway and likely occurs by a lateral translocation in the plane of the ER membrane [66]. Importantly, TGB3 is able to direct TGB2 from intracellular ER-associated sites to PAMBs [39], although in some experiments TGB2 is found in PAMBs even when expressed in the absence of TGB3 [68]. The observed TGB3 ability to change the subcellular localization of TGB2 suggests interaction between TGB2 and TGB3. Indeed, residues involved in such interactions have been mapped in both proteins of BSMV, and mutations affecting these residues inhibited virus transport [55].

Virus cell-to-cell movement is generally accepted to require a modification of the PD internal structure resulting in the PD SEL increase [14]. For potex-like TGB proteins, attempts to identify the TGB protein responsible for the PD SEL increase have resulted in contradictory data. Several experiments have shown that potex-like TGB1 introduced into cells by microinjection is able to interact with PD and increase the PD permeability [69,70,71]. In other studies, only TGB2 but not TGB1, TGB3, or CP is shown to be able to increase the PD SEL [72]. The hordei-like *Potato mop-top virus* (PMTV) TGB2 protein is shown to interact with cellular proteins TIP1, TIP2 и TIP3, which in turn can interact with β-1,3-glucanase and therefore influence the callose degradation at the PD neck regions, resulting in the increase of the PD SEL [73].

### 2.2. Early Model of TGB-Mediated Transport

An initial model of TGB functioning was based on experiments employing transient co-expression of individual cloned TGB proteins for analysis of their intracellular and intercellular transport. In a representative study, the PMTV TGB2 and TGB3 proteins have been shown to act in cooperation to ensure delivery of TGB1 to TGB2/TGB3-containing PAMBs at the PD entrance. This delivery does not require the TGB1 NTPase/helicase activity, which, instead, is essential for further TGB1 transport into the PD interior and to neighboring cells [43]. These and other experiments have demonstrated that TGB2 and TGB3 function in cells where these proteins have been synthesized and are not transported through the PD channels into neighboring cells. As TGB1 proteins are generally assumed to bind viral RNA or virions to produce transport-competent genome-containing vRNPs [39,63], the following model for TGB-mediated virus cell-to-cell movement has been proposed. The TGB1-containing vRNPs, which can be either TGB1:RNA complexes or TGB1-bound virions, are transported intracellularly to PD due to (i) interaction of TGB1 with TGB2/TGB3 and (ii) signals in TGB2 and/or TGB3 directing these proteins to PAMBs at the PD orifice. Further transport of viral genomes through PD depends on the enzymatic activities of TGB1, presumably required to remodel either the structure of the transported vRNP, the internal PD structure, or both [39,43]. This generalized model, although correct in describing the functions of individual proteins, appeared incomplete and has been further considerably developed in view of recently accumulated data on (i) direct and tight links between virus cell-to-cell movement and replication and (ii) the pivotal role of MP-mediated reorganization of cell membranes for virus transport. 

### 2.3. The Role of Modified ER Membranes in TGB-Mediated Transport

Fine details of coupling virus transport to replication in ER-derived membrane compartments are being studied using potexviruses as models, and the current knowledge can be summarized as follows.

In the cytoplasm, the virus replicase molecules form high-molecular-weight complexes [74] targeted to the ER membrane by an amphipathic α-helix and form, in the absence of other virus proteins, initial viral replicative compartments (iVRCs), which are clustered in the perinuclear area into large amorphous aggregates [75] morphologically similar to structures found in potexvirus-infected cells and historically called ‘X-bodies’ [68,76]. The X-bodies contain all PVX-encoded proteins, virions, and host factors. Detailed analysis of X-bodies revealed that they represent complex layered structures. The TGB1 protein plays an important role in their organization, as TGB1 aggregates form the core of the X-body, or so-called ‘beaded sheets’, visible by electron microscopy in infected cells. Additionally, TGB1 reorganizes the actin cytoskeleton at the X-bodies [77,78]. The TGB1 aggregates are surrounded by recruited TGB2/TGB3-containing ER membranes representing a highly dense reticulated network rather than a native polygonal tubular ER network [78]. Analysis of the localization of the viral dsRNA in the X-bodies demonstrate that RdRp/dsRNA-containing bodies surround TGB1 aggregates, are covered by ‘chain mail-like’ TGB2-aggregates, and localize in the immediate vicinity to TGB3 aggregates [61]. Non-encapsidated viral RNA and progeny virions localize to the periphery of the X-bodies [61]. Therefore, the X-bodies represent highly-structured stationary virus factories associated with the ER. 

At early infection stages, PVX TGB2 induces the formation of tiny mobile ER-associated membrane bodies [79]. In virus-infected cells, these bodies also contain the viral replicase [63,79]. The formation of the latter bodies likely results from the direct interaction of TGB2 with the viral replicase (Figure 1). Regions involved in this interaction have been mapped in both proteins and the TGB2 incorporation into the replicase containing structures does not require other TGB proteins [61]. Therefore, the TGB2 protein is suggested to target iVRCs [61] to convert them into movement VRCs (mVRCs) capable of intracellular transport. TGB3 also incorporates mVRCs, likely due to its interaction with TGB2 [80], whereas the TGB1 recruitment requires both TGB2 and TGB3 [68]. Thus, iVRCs can form stationary replicase-containing perinuclear X-bodies even in the absence of other viral proteins and, additionally, can develop into mobile mVRCs in the presence of the TGB proteins.

The formed mVRCs are believed to be highly mobile and can be transported along the cytoskeleton to the PD entrance. In fact, TGB2-containing membrane bodies have been shown to colocalize with microfilaments (MF) and an intact actin cytoskeleton is required for the efficient cell-to-cell transport of PVX [79,81]. Thus, the signals of intracellular trafficking identified in TGB3 (see above) and presumed for TGB2 [68] can be required for the PD targeting of TGB3-containing mVRCs rather than the TGB proteins on their own. At the PD entrance, mVRCs form PAMBs, which represent ER-derived PD-anchored sites where virus replication is coupled to cell-to-cell virus transport [68] (Figure 1). Importantly, PAMBs contain reorganized densely reticulated ER tubules, which likely comprise the backbone of these structures [68]. Therefore, the ER membranes in PAMBs and X-bodies are modified in a similar manner, emphasizing the structural similarity of both types of iVRC-derived compartments.

Once vRNP complexes are synthesized in PAMBs, they are immediately directed into the PD channel by the TGB1 protein for further transport into neighboring cells [68]. Viral MPs studied so far demonstrate nonspecific RNA binding activity, suggesting that MPs are not able to distinguish between vRNAs and cellular mRNAs [82]. Therefore, besides physical concentration of necessary viral and host factors, PAMBs may possibly increase the specificity and efficiency of viral RNA transport by excluding cellular RNAs [76].

Therefore, the available data suggest that TGB2 appears to play the central role in the organization of PAMBs, as it: (1) directly interacts with the viral RdRp and therefore targets iVRCs and converts them into mVRC; (2) interacts with TGB3, which contains signals of PD targeting that facilitates the mVRC transport to the PD orifice; (3) together with TGB3 recruits TGB1 to mVRCs. 

### 2.4. BMB2 Interaction with ER Membranes

New light is shed on the TGB2 functions in studies of MPs encoded by *Hibiscus green spot virus* (HGSV; family *Kitaviridae*), which has a transport gene module evolutionary related to TGB [40,83,84]. This transport module is called ‘binary movement block’ (BMB) as it consists of two open reading frames, accordingly called BMB1 and BMB2 genes [85]. The virus movement function of the BMB1 and BMB2 proteins has been confirmed in complementation experiments, where these two HGSV proteins complement cell-to-cell transport of the modified PVX genome, which is able to replicate and express GFP in initially infected cells, but is deficient in cell-to-cell movement because of the absence of TGB and CP genes [85].

The BMB1 protein is distantly related to the TGB1 protein in sequence and contains the NTPase/helicase domain [40,84]. Although not studied experimentally, BMB1, by analogy to TGB1, is suggested to be the RNA-binding protein that takes part in the formation of transport vRNPs. This view is indirectly confirmed in the PVX complementation experiments mentioned above, as in this system the BMB1 protein is the only viral protein able to form transport vRNPs [85].

The BMB2 protein, similar to TGB2, has two highly hydrophobic transmembrane domains, with the central region between them showing limited sequence similarity to TGB2 proteins [62,86]. BMB2 is integrated into the ER membrane and is localized in the ER-derived PAMBs formed upon BMB2 expression [85]. The intracellular BMB2 transport to PAMBs does not involve the Golgi-dependent secretory pathway, as earlier shown for TGB2, but requires the intact actin/ER network, suggesting that both BMB2 and TGB2 can employ a similar mechanism of transport to PAMBs, likely involving the lateral translocation in the plane of the ER membrane [87]. In transient co-expression experiments, BMB2 directs BMB1, which exhibits a diffuse localization when expressed alone, to PAMBs, to the PD interior, and through the PD channels to neighboring cells [85]. Therefore, as the HGSV transport system lacks a counterpart of TGB3, the functions of both TGB2 and TGB2 proteins can be apparently performed by the single BMB2 protein. 

Recently, the BMB2 protein has been shown to have reticulon-like properties that provided a clue to its functions, as well as those of TGB2, in virus cell-to-cell transport [85]. Reticulons are cellular proteins essential for the generation of the ER membrane curvature and for the formation of ER tubules, serving therefore for the formation and maintenance of the ER morphology [88,89,90]. The reticulon functions are determined by a ‘reticulon homology domain’ comprising two membrane-immersed hydrophobic regions forming wedge-shaped hairpins, which bend the lipid bilayer as the reticulon oligomerizes [91,92]. These regions are connected by a loop, and therefore the reticulon molecule has a W-like topology in the ER membrane, where both ends of the protein and the central loop are located on the ER cytosolic side [91]. The BMB2 protein, similar to the reticulons, has been shown to have the W-like topology in the membrane, and the overexpression of BMB2 results in the constriction of the ER tubules [93], as previously demonstrated in similar experiments with reticulons [91]. Therefore, BMB2 is supposed to generate, either on its own or by interaction with cellular protein partner(s), the ER membrane curvature. Importantly, the PVX TGB2 protein, when transiently expressed in plant cells, is also able to induce constrictions of ER tubules, and similar constrictions have been observed in PVX-infected cells [93]. Mapping of PVX TGB2 regions interacting with virus RdRp shows that the central and C-terminal protein regions are located in the cytoplasm [61], implying that TGB2 can also adopt the W-shaped conformation in the ER membrane. Thus, the formation of aberrant densely reticulated ER tubules observed on PAMBs and X-bodies in PVX-infected cells can be attributed to the presumed TGB2 ability to modify the structure of ER tubules. As the modified ER membranes are considered a core structural element of VRCs/PAMBs/X-bodies, the modification of ER tubules by the BMB2/TGB2 proteins can be essential for the formation of initial VRCs, which are further developed into PAMBs or X-bodies upon recruitment of other viral proteins. This model is corroborated by observations that the formation of membrane replicative compartments of a variety of (+)RNA viruses replicating in the cytoplasm requires the reticulon activity, and viruses without own reticulon-like proteins recruit cellular reticulons for membrane modification [94].

Another characteristic BMB2 feature shared with reticulons is the affinity to highly curved membranes. In cells containing flat ER cisternae, both reticulons and BMB2 are targeted to cisternae rims with high membrane curvature [91,93]. These findings can explain the observed localization of BMB2 and TGB2 in the PD channel or in the PD central cavity [68,93], as these proteins may have affinity to desmotubule, the ER tubule on the PD axis that has a small diameter and, consequently, a high membrane curvature. Such BMB2/TGB2 localization makes a parallel with two plant reticulons, RTNLB3 and RTNLB6, found in the desmotubule [95,96] and, on the other hand, can be pertinent to the observed ability of BMB2 and TGB2 to increase the PD SEL [72,93]. The mechanism of BMB2/TGB2-induced SEL increase is unknown. However, taking into account the BMB2 ability to form high-molecular-weight complexes likely including BMB2 oligomers, the BMB2 protein has been hypothesized to displace, upon accumulation of such complexes in the desmotubule, native components of the PD channel and therefore increase the PD SEL [93].

Thus, two key functions of BMB2/TGB2, namely the modification of ER membranes for VRC formation and the modification of the PD internal structure resulting in the PD SEL increase, may be determined by the mode of BMB2/TGB2 interaction with the ER membrane, involving both the generation of additional membrane curvature and the targeting to membrane sites with a high curvature of the lipid bilayer.

## 3. Potyviruses: The Vesicles/Endosomes-Based Pathway

### 3.1. Potyvirus Proteins Involved in Virus Transport

Genomes of potyviruses (genus *Potyvirus*, family *Potyviridae*) encode a polyprotein precursor, which is autocatalytically processed into eleven mature proteins [97]. In addition to polyprotein-derived proteins, a 7 kDa protein named PIPO is produced in infected cells. Due to RNA polymerase slippage during the replication, an additional adenine residue is inserted in 1–2% of genomic RNA progeny that brings the PIPO-encoding sequence into the reading frame of another viral protein called P3. As a result, a fusion protein P3N-PIPO consisting of the amino-terminal half of P3 and the PIPO polypeptide is produced [98,99]. 

Potyviruses have a multicomponent transport system, as several viral proteins are involved in potyvirus cell-to-cell transport, including CP, P3N-PIPO, and the cylindrical inclusion protein (CI). Potyvirus mutants encoding encapsidation-deficient CP are incapable of cell-to-cell movement [100,101,102], indicating a direct link between virion formation and virus transport. Moreover, the potyvirus CP has been found in the PD interior by immunogold labeling, being detected in the form of fibrils morphologically similar to flexuous filamentous potyvirus virions [103,104]. Therefore, potyviruses are suggested to move cell-to-cell in the form of either genuine virions or virion-like complexes similar to virions in their composition but differing in their fine structure [105,106]. CI, another protein essential for potyvirus transport [103,104,107], contains a helicase domain and exhibits the NTPase and helicase activities involved in virus replication [108]. Mutations in the N-terminal protein region, which is required for the CI self-interaction and is positioned outside the helicase domain, abolish virus movement but do not block replication [107,109,110], suggesting that these two CI functions can be uncoupled. In potyvirus-infected cells, the CI protein forms pinwheel cytoplasmic inclusions, which are located at the PD entrance and often traverse the cell wall over the PD channels, and numerous virions are typically associated with the CI-formed structures [103,104]. When expressed alone in plant cells, CI is found in aggregates in the cytoplasm. However, upon co-expression with P3N-PIPO, CI forms inclusions similar to those in potyvirus-infected cells, showing that the CI transport to PD is P3N-PIPO-dependent [111]. In fact, P3N-PIPO on its own is targeted to PD and interacts with CI in plant cells [111]. Similar to the TMV MP, the P3N-PIPO protein is able to move cell-to-cell in plant tissue and increases the PD SEL [98]. P3N-PIPO binds PCaP1, a cellular cation-binding protein interacting with the PM via myristoylation. Moreover, the potyvirus transport, but not replication, is drastically reduced in PCaP1-deficient *Arabidopsis* plants [98]. These data demonstrate that the PCaP1-dependent targeting of P3N-PIPO to the PM is essential for virus transport.

Two additional potyvirus-encoded proteins implicated in cell-to-cell transport, VPg and HC-Pro, interact with each other [112,113]. The VPg protein is covalently linked to the 5′-terminus of potyvirus genomic RNA, and the VPg interaction with HC-Pro is suggested to be responsible for the formation of a morphologically distinct structure on one end of potyvirus virions that can be labeled with HC-Pro antibodies [114,115]. The CI protein is co-purified with a subpopulation of potyvirus particles and associated, according to immunogold labeling, with one end of the virion [116], likely due to the reported interaction of CI with HC-Pro [113]. 

Collectively, the available data suggest that a complex pattern of interactions between potyvirus-encoded proteins is required for virus cell-to-cell transport. According to the proposed model [53,105], which implies that the virion is the most probable transport form of the potyvirus genome, a fraction of progeny virions may have a terminal structure consisting of VPg, HC-Pro, and CI that may make these virions competent for cell-to-cell transport. The targeting of such virions to PD is possible as a result of the interaction between the virion-associated CI and P3N-PIPO, which is capable of localization to PD, and the CI self-interaction involved also in the formation of the PD-associated pinwheel inclusions. Further cell-to-cell transport of virions likely depends on the PD SEL increase induced by P3N-PIPO and, additionally, HC-Pro and CP [117]. However, this mechanistic model does not take into account the association of potyvirus replication with membranous structures and the pathways of intracellular targeting of movement-related proteins to PD.

### 3.2. Formation of Potyvirus VRCs

The replication of potyviruses is associated with the EMS membranes, as also documented for many other animal and plant RNA viruses replicating in the cytoplasm [118,119]. Specifically, the potyvirus replication occurs in vesicles derived from the ER membranes [120,121] (Figure 1). These vesicular VRCs carry viral proteins including CI and the RNA-dependent RNA polymerase NIb, as well as virus RNA and cellular factors [120,122,123]. However, the fine VRC structure is unstudied. In particular, it remains unclear whether the virus-specific components are located in the VRC lumen or on the cytoplasmic face of the VRC-enclosing membrane.

The biogenesis of potyvirus ER-derived VRCs depends on the viral protein termed 6K_2_, a small protein with a single transmembrane domain cotranslationally integrated into the ER membrane and capable of inducing the VRC formation in the absence of other viral proteins [120]. In the context of potyvirus infection, the key player in VRC formation is the 6K_2_-VPg-Pro precursor serving as a scaffold, the VPg moiety of which recruits several virus and host proteins involved in replication, whereas the 6K_2_ moiety is responsible for the membrane remodeling [124]. In fact, the 6K_2_-dependent reorganization of ER in potyvirus-infected cells leads to the formation of two types of structures, namely motile VRCs located at the cell periphery and a large perinuclear globular structure, which includes the ER and Golgi structures as well as chloroplasts [125]. However, the 6K_2_-induced structures are not functionally isolated, as the motile VRCs can derive from the perinuclear structure [125]. 

The 6K_2_-induced formation of vesicular VRCs is COPII-dependent [126], i.e., it involves the conventional cellular mechanism of COPII (coat protein complex II)-mediated formation of transport vesicles, which bud from the ER membrane and deliver cargo to the Golgi cisternae [5]. Accordingly, 6K_2_ is found in the ER exit sites, the discrete ER domains where secretory proteins and membrane proteins destined for the export from the ER are concentrated and packed into transport vesicles [126,127]. Furthermore, 6K_2_ interacts with the COPII coatomer Sec24a, the subunit interacting with cargo proteins for their packaging into COPII-vesicles [128]. The N-terminal cytoplasmic domain of 6K_2_ is essential for this interaction, and a point mutation of a conserved residue in this domain leads to partial retention of the protein in the ER and blocks the virus transport out of primary infected cells [128]. Therefore, the formation of potyvirus VRCs occurs upon the 6K_2_-induced COPII-dependent detachment of 6K_2_-containing vesicles from the ER. Interestingly, in cells coinfected with two fluorescently tagged versions of one potyvirus genome modified to express either GFP or mCherry fused to 6K_2_, a part of VRCs have been observed to carry green-only or red-only fluorescent labeling, suggesting that a particular individual VRC derives from a single genome and likely includes locally synthesized viral proteins [122].

In potyvirus-infected cells, the export of cellular proteins from the ER is inhibited, possibly due to the Sec24a mobilization by the 6K_2_ protein [125]. In *Arabidopsis* mutant lines with partially functional Sec24a, the inhibition of COPII-dependent exit from the ER is accompanied by the formation of aberrant membrane compartments somewhat similar to large perinuclear globular structures observed upon potyvirus infection [124]. These findings suggest that the formation of both discrete motile VRCs and perinuclear structures may be induced through a common mechanism involving Sec24a. 

Based on available data, we conclude that the VRC nature and biogenesis differ drastically between TGB/BMB and the potyvirus transport system (Figure 1). While the TGB/BMB VRCs are structurally based on modified, likely additionally constricted, ER tubules and remain a part of the ER, the potyvirus VRCs are vesicular structures detached from the ER membrane by use of the cellular COPII machinery. Additionally, the TGB/BMB VRCs are stationarily located at the PD entrance [68], whereas the potyvirus VRCs represent motile membrane structures. 

### 3.3. VRC Trafficking and the Endosome/Post-Golgi Pathway

The motility of potyvirus VRCs in the cytoplasm requires the actin network but not microtubules, and the VRC trafficking along actin microfilaments depends on myosins XI-2 and XI-K [122,125,126,129], suggesting a mechanism described for cellular motile endomembrane vesicles associated with F-actin [130]. In potyvirus-infected cells, the VRCs are found in association with the PM, often being located in close vicinity of PD [131], suggesting that VRCs can reach PD as a result of their intracellular movement. Moreover, using a photoactivated fluorescent protein to label 6K_2_, the potyvirus VRCs are shown to move cell to cell [131], demonstrating therefore the possibility of intercellular transport of virus replication membrane compartments, as it has been suggested for the TMV VRCs [132]. 

The VRC intracellular transport is routed by the 6K_2_ protein, as mutations of conserved glycine residues in the 6K_2_ transmembrane domain abolish the normal VRC trafficking and redirect VRCs to the Golgi structures that block the virus replication and transport [133], indicating that the normal VRC transport bypasses the Golgi structures and uses a specific nonconventional pathway (Figure 1). The RHD3 (ROOT HAIR DEFECTIVE 3) protein, a cellular dynamin-like atlastin, is mobilized to the potyvirus VRCs [134]. A potyvirus mutant carrying the substitutions of glycine residues in the 6K_2_ transmembrane domain is incapable of the RHD3 recruiting to VRCs and deficient in the replication and movement. Additionally, the potyvirus infection is suppressed in *rhd3* mutant plants [134], suggesting that RHD3 may be required for the 6K_2_ function in the proper VRC routing. 

The Golgi-independent VRC transport can involve post-Golgi membrane compartments. In fact, the 6K_2_ protein is colocalized and copurified with VTI11 [133], a Qb-SNARE protein localized to the trans-Golgi network and late endosome/multivesicular bodies (MVBs) [135]. In line with this finding, double-stranded RNA representing a replication intermediate of potyvirus genome is detected in MVBs by immunogold electron microscopy [133]. Moreover, an *Arabidopsis* VTI11 knock-out line is resistant to potyvirus infection [133], showing the importance of VRC targeting to MVB. This targeting depends on a specific interaction of the 6K_2_ protein with VSR4, an ESCRT protein, and a point mutation in VSR4 leads to resistance to potyvirus infection [136,137]. Therefore, the intracellular transport of potyvirus VRCs, which are detached from the ER and finally reach PD, can bypass the Golgi and involve post-Golgi membrane compartments (Figure 1). Interestingly, *Plantago asiatica mosaic virus*, a TGB-containing potexvirus, is capable of establishing systemic infection in the VTI11 knock-out plants resistant to potyvirus infection [133], emphasizing the difference between membrane-associated replication/transport systems of potyviruses and viruses with TGB/BMB.

Another role of post-Golgi membranes in the potyvirus transport is demonstrated in studies employing RabE1d(NI), a dominant negative mutant of the small GTPase RabE1d involved in the post-Golgi trafficking toward the PM. RabE1d(NI) has no effect on the virus replication and the VRC intracellular motility, but blocks the VRC cell-to-cell transport [129,138]. This RabE1d(NI) effect appeared to be indirect, as the P3N-PIPO localization to PD is suppressed in RabE1d(NI)-expressing cells that leads to blocked VRC targeting to PD [138].

The transport of 6K_2_-induced vesicles can be reconstructed in non-infected cells where individual potyvirus proteins are transiently co-expressed. Under these conditions, 6K_2_ induces the formation of vesicles morphologically indistinguishable from typical VRCs observed in infected cells, whereas the CI and P3N-PIPO proteins are necessary and sufficient for cell-to-cell transport of these vesicles [138]. The 6K_2_ protein interacts with CI, and such interaction is required for the intercellular movement of 6K_2_ vesicles [138]. These and earlier findings elucidate the specific roles played by P3N-PIPO, CI, and 6K_2_ in the VRC trafficking. P3N-PIPO is localized to PD, modifies the PD internal structure, and increases the PD SEL; the CI protein is targeted to PD due to its interaction with P3N-PIPO; in turn, the VRCs are targeted to the CI-formed pinwheel inclusions at the PD entrance due to the interaction of 6K_2_ with CI. Indeed, the 6K_2_-induced vesicles are observed in close association with PD and the CI-formed PD-associated inclusions [131,138].

Therefore, two models, both supported by experimental data, are currently suggested for the potyvirus cell-to-cell movement: the model of virion transport outlined above and the more recent model of VRC cell-to-cell transport. Two key protein components of the potyvirus transport system, as well as their proposed functions, are common for both models, i.e., (i) P3N-PIPO, which targets PD, and (ii) the CI protein, which is directed to PD due to its interaction with P3N-PIPO and forms characteristic PD-associated inclusions. The genome-containing infectious entities transported through the PD channels are, however, different in the two models, being either modified virions carrying CI at one end and therefore capable of interaction with P3N-PIPO and/or CI molecules forming inclusions, or VRCs carrying 6K_2_ and therefore interacting with CI and the pinwheel inclusions. At the moment, it is not clear how these two models can converge, as the VRC model does not explain why the potyvirus CP is essential for virus cell-to-cell movement, while the virion model does not consider the observed intercellular transport of 6K_2_-containing vesicles. 

Irrespective of the nature of a potyvirus infectious entity transported through the PD channels, recent findings showing that entering the endosome/post-Golgi pathway is essential for a potyvirus infection making parallels with the mechanism of intracellular transport discovered for cellular mRNAs.

## 4. Endosome Structures in mRNA Transport

Generally, the intracellular transport of mRNA is required for specific localization of particular mRNA in eukaryotic cells that ensures the localized protein synthesis essential for establishing cell polarity, patterning, and cell fate determination [139]. Additionally, the mRNA transport is indispensable in polarized cells, such as mammalian neurons, in which distal dendrites and axons spatially separated from the nucleus-containing cell body are provided with transcripts by means of the mRNA transport [140] or cells of distal fast-growing tips of fungal hyphae, in which the mRNA delivery from the nucleus is required for the localized synthesis of protein components essential for the growth [141]. 

Although the mRNA transport has been well characterized in several non-plant systems (see below), the studies of mRNA transport mechanisms in plants are in their very beginning. Currently, the main model for such studies is the mRNA transport in endosperm cells of developing rice seeds, where mRNAs encoding the storage proteins glutelin and prolamine are transported to two distinct ER-associated endomembrane compartments [142]. For their differential localization, the two mRNAs contain *cis*-acting signals, which are termed zip codes. In the prolamine mRNA, two zip-code elements with a common sequence motif are located in the protein coding region and the 3′-UTR, whereas the glutelin RNA contains six zip-code elements of two types found in the coding region (five such elements) and the 3′-UTR [143]. Several RNA-binding proteins (RBPs) interact with the zip codes to form transport RNPs. The key role is played by the RBP-P/RBP-L complex, which specifically binds zip codes and serves as a scaffold for interaction with other RNP components [144]. In endosperm cells, the glutelin and prolamine RNAs are detected as particles moving in the cytoplasm [145,146]. However, the nature of these particles remained unclear until recently, when other proteins interacting with the RNPs have been discovered. In particular, the RBP-P/RBP-L complex has been shown to interact with an N-ethylmaleimide-sensitive factor (NSF) and the small GTPase Rab5a, the two proteins participating in endosomal membrane trafficking; moreover, the RBP-P/RBP-L proteins colocalize with NSF and Rab5a on endosomes carrying the glutelin mRNA [147]. In Rab5-deficient plants, the glutelin mRNA-containing RNPs are mistargeted to extracellular structures as a result of corrupted endosome trafficking [147]. Collectively, these findings demonstrate that mRNAs can be transported in plant cells in the form of RNPs anchored to the surface of endosomes, making evident parallels with the potyvirus-specific transport pathway involving endosomes. This mechanism of mRNA transport is not unique for plants and the role of endosomes in mRNA transport is well documented for other systems, including fungal hyphae and animal axons. 

Similar to rice endosperm cells, the endosome-dependent intracellular mRNA trafficking in the filamentous forms of the fungus *Ustilago maydis* requires a number of proteins involved in transport RNP formation and its anchoring to the endosome surface [141,148]. In particular, the RNPs are formed with the participation of Rrm4 and Grp1, two RNA-binding proteins interacting predominantly with the 3′-UTRs of shared cargo mRNAs, often in close proximity, and Pab1, a poly(A)-binding protein [148,149]. The transport RNPs are anchored to endosomes by UPA1, a protein that interacts with Pab1 and, on the other hand, contains the FYVE zinc finger domain enabling specific interaction with lipids on the endosome surface [150]. The transport RNP-carrying endosomes move along the microtubule by the use of kinesin and dynein motors [151]. Importantly, translationally active polyribosomes are efficiently co-transported with endosomes in *U. maydis* [152,153], suggesting that the endosomal transport system enables translocation of functional translation apparatus that ensures the immediate local protein synthesis upon mRNA delivery to growing distal hyphae ends. 

Neuronal axons are often extremely long and can extend for millimeters. To avoid the prolonged time of response to external stimuli that would drastically slow down the axon functioning, neurons employ rapid mRNA trafficking from the nucleus to distal cell parts and localized protein synthesis as the ways to overcome the distance constraints [154,155]. In the axons of *Xenopus laevis* retinal ganglion cells, 20–30% of mRNAs colocalize with endosomes displaying bidirectional intracellular transport, and a number of ribosomal proteins and RBPs, such as Fragile X mental retardation protein, Staufen2, and Pumilio2, are associated with both mRNAs and endosomes, suggesting that these organelles could act as mRNA transport platforms [156]. In mice models, the localized translation of β-actin in axons is critical to the process of growth cone pathfinding [157]. For selective intracellular transport and localized translation, the neuron β-actin mRNA is targeted to late endosomes together with ribosomes and the RNA-binding protein ZBP1 that binds to a 54-nucleotide zip code sequence in the 3′-UTR of β-actin mRNA [158,159]. Similar to the β-actin mRNA, mRNA of GAP-43, another protein selectively localized to axonal growth cones during development, localizes in axons in a ZBP1-dependent manner [158,160]. Recently, FERRY, a novel early endosome-transported RNP complex composed of five subunits called Fy-1 to Fy-5, has been identified in human neurons. The small GTPase Rab5 binds both early endosomes and Fy-2, therefore tethering the FERRY RNP on moving membrane organelles [161]. 

The above examples demonstrate that the endosome-dependent mRNA trafficking is observed in plant, fungal, and animal cells, therefore the involvement of endosomes in the potyvirus cell-to-cell transport does not appear somewhat uncommon. However, the TGB/BMB-specific transport pathway has no parallels among currently known cellular mechanisms of RNA transport.

## 5. Conclusions

Here, using the TGB/BMB-mediated virus transport and the potyvirus-specific transport as examples, we substantiate the idea that plant viruses can use dissimilar, unrelated mechanisms of modification of cell endomembranes for their movement in plants. In particular, the proteins of the TGB/BMB transport system modify the ER tubules, thereby generating movement-related replication compartments that remain a part of the ER and are stationarily located at the PD entrance, so that the genomic RNA progeny is directed into the PD microchannels for cell-to-cell transport. By contrast, the potyvirus-specific replication/movement membrane compartments represent motile vesicular structures detached from the ER membrane, and the trafficking of these structures to PD involves targeting to post-Golgi compartments, including endosomes, which appear to be an essential hub for the potyvirus infection. Moreover, vesicular membrane structures carrying virus proteins and genomic RNA are suggested to be transported through PD to enable virus cell-to-cell transport. Therefore, according to the different ways of interaction with endomembranes, the mechanisms of transport, both intracellular and intercellular, appear to be fundamentally different between potyviruses and TGB/BMB-encoding viruses. Undoubtedly, detailed analyses of host membrane modifications observed for plant virus transport systems of other types may either allow their assignment to one of the transport mechanisms discussed in this review, or lead to the discovery of yet unknown mechanisms of endomembrane targeting and cell-to-cell transport. For example, studies of ‘double gene block’ (DGB), the transport system of carmoviruses, demonstrate that the intracellular trafficking of a DGB-encoded protein involves Golgi [162], showing a marked difference from both transport systems discussed in this review and suggesting that DGB may use a distinct specific transport pathway. In conclusion, we emphasize that studies of plant virus interaction with endomembranes are of key importance for understanding the mechanisms of virus cell-to-cell transport.

## Figures and Tables

**Figure 1 plants-11-02403-f001:**
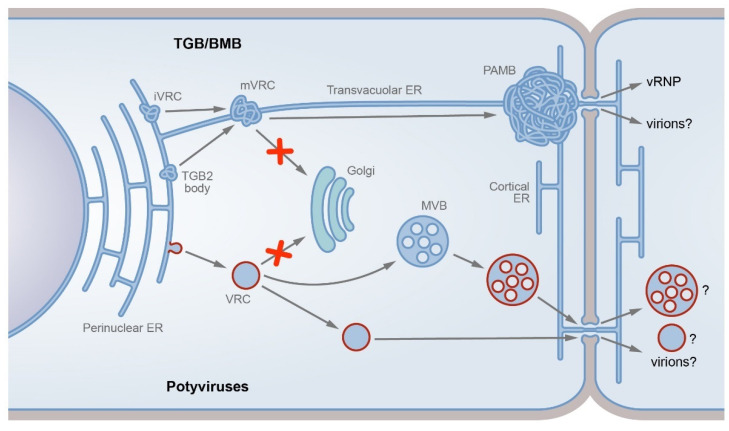
TGB/BMB-specific and potyvirus-specific transport pathways. The two dissimilar pathways described are either modification of the ER tubules (TGB/BMB; upper part) or formation of motile vesicles detached from the endoplasmic reticulum and targeted to endosomes (potyviruses, bottom). In the TGB/BMB-specific pathway, the viral replicase forms initial virus replication complexes (iVRCs) on the ER membrane, whereas TGB2 can interact with the ER to form membrane TGB2 bodies. Due to TGB2 interaction with viral replicase, TGB2 targets iVRCs to convert them into movement VRCs (mVRCs). The mVRCs are capable of intracellular transport and can be transported along the cytoskeleton to the PD entrance, bypassing the Golgi-dependent trafficking pathway (shown by red cross sign). At the PD entrance, mVRCs form PD-associated membrane bodies (PAMBs), which represent ER-derived PD-anchored sites where virus replication is coupled to cell-to-cell virus transport. Virus ribonucleoprotein complexes (vRNPs) containing genomic RNA and MPs are formed in PAMBs and directed to the PD channels. Alternatively, virions, likely MP-modified, can be the virus transport form (indicated by question mark). In the potyvirus-specific pathway, the viral 6K_2_ protein induces the formation of VRCs on the ER membrane. The potyvirus VRCs, which carry viral proteins, virus RNA, and cellular factors, are detached from the ER membrane in a COPII-dependent manner. Further VRC intracellular trafficking bypasses the Golgi pathway (red cross sign). The VRCs target late endosomes (multivesicular bodies, MVBs), which in turn target PD. Alternatively, the VRCs can directly target PD. The potyvirus-specific entity transported through the PD channels is unknown and can be VRC, modified MVB, or virion (indicated by question marks).

## Data Availability

Not applicable.

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
