# Peer review of "Distinct Mechanisms of Endomembrane Reorganization Determine Dissimilar Transport Pathways in Plant RNA Viruses"

_plants, 2022, doi:10.3390/plants11182403_

Round 1

Reviewer 1 Report

Solovyev et al îs well organized and informative. However, inclusion of phloem movement of +ve RNA virus families such as Luteoviridae, closteroviridae can increase the scope of the review.

Line 39: Expand PM

Line 267: mention family of hgsv

Author Response

Solovyev et al is well organized and informative. However, inclusion of phloem movement of the RNA virus families such as Luteoviridae, closteroviridae can increase the scope of the review.

In this review, we focus mainly on the intracellular transport and associated membrane modifications and therefore intentionally leave out of review the phloem virus transport, for which association with membranes is not evident. For this reason, in the revised version we stay within the originally defined scope of the paper.

Line 39: Expand PM

Line 267: mention family of hgsv

Done.

Reviewer 2 Report

This manuscript is a thorough review regarding the mechanism of plant viral movement involving the TGB/BMB transport system and the potyvirus-specific transport. The manuscript provides in-depth information to understand the mode of plant virus movement mediated by two unrelated mechanism. The manuscript is well prepared and I recommend it for publication. I have several suggestions (added in the pdf file).

Author Response

I have several suggestions (added in the pdf file).

We agree with many suggested changes and have modified the text accordingly.

The comment we disagree with is:

need to further describe the interactions among TGB1, TGB2 and TGB3 as reported by previous studies

In this paragraph, only the interaction between TGB2 and TGB3 is discussed. Such interaction is mapped only in the BSMV proteins, which are already mentioned in this paragraph with an appropriate citation.

Reviewer 3 Report

 The manuscript provides a detailed review of the intracellular and intercellular transport of an important group of viruses in host plants. The review's primary focus is the group with (+)RNA-genomes. The review starts with plants' typical inter and intracellular transport (independent of viral infections) and describes the plant endomembrane system (EMS). It then compares two types of interaction virus-plant endomembranes: 1) based on the modification of the endoplasmic reticulum tubules, or 2)Formation of motile vesicles detached from the endoplasmic reticulum and targeted to endosomes. Further, the review draws parallels between virus-specific RNAs' intracellular transport mechanisms and cellular mRNAs.

The main suggestion to authors is to clarify the ideas in the manuscript: the numerous (and pertinent) citations hide the conclusions. This style makes the reading difficult.

Title: The review does not cite plant viruses in general and how the mechanisms described can be applied to them. For this reason, the title "Dissimilar pathways of endomembrane system-dependent plant virus movement" is inappropriate. The title could include the following information: Distinct mechanisms of endomembrane reorganization lead to dissimilar virus transport pathways in RNA+ viruses.

Abstract: Instead of: "Here, we discuss two plant virus transport systems, the potyviral one, on the one hand, and the MPs encoded by the triple gene block and the binary movement block, two related transport gene modules, on the other hand. These transport systems use unrelated mechanisms of endomembrane reorganization and, accordingly, different virus transport pathways, which are based on either modification of the endoplasmic reticulum tubules, or formation of motile vesicles detached from the endoplasmic reticulum and targeted to endosomes. Based on the presented considerations, we emphasize that the mechanism of plant virus cell-to-cell transport is determined by the mode of virus interaction with cell endomembranes" I suggest: "Here, we discuss two distinct virus transport pathways based on either modification of the endoplasmic reticulum tubules or formation of motile vesicles detached from the endoplasmic reticulum and targeted to endosomes. The viruses with the movement proteins encoded by the triple gene block exemplify the first, and the potyviral system is the example of the second type. These transport systems use unrelated mechanisms of endomembrane reorganization. We emphasize that the mode of virus interaction with cell endomembranes determines the mechanism of plant virus cell-to-cell transport."

Figure 01 is a very nice summary of the main idea of the review. 

The figure 01 legend should not simply state "See text for details" but take advantage of the very nice visual tool to describe and summarize the review. 

Red signs, question marks, and arrows should summarize the central idea of ​​the manuscript.

I suggest adding: "The two dissimilar pathways described are either modification of the endoplasmic reticulum tubules - TGB/BMB-specific (upper part) or formation of motile vesicles detached from the endoplasmic reticulum and targeted to endosomes - potyvirus-specific (bottom). 

After this introduction, add a brief step-by-step description of each mechanism.

Conclusions:

"Here, using the TGB/BMB-mediated virus transport and the potyvirus-specific transport as examples, we substantiate the idea that plant viruses can use dissimilar, unrelated mechanisms of modification of cell endomembranes for their movement in plants." 

Show which virus groups these two described groups are representative of and why.

Author Response

Title: The review does not cite plant viruses in general and how the mechanisms described can be applied to them. For this reason, the title "Dissimilar pathways of endomembrane system-dependent plant virus movement" is inappropriate. The title could include the following information: Distinct mechanisms of endomembrane reorganization lead to dissimilar virus transport pathways in RNA+ viruses.

The title is changed as suggested.

Abstract: Instead of: "Here, we discuss two plant virus transport systems, the potyviral one, on the one hand, and the MPs encoded by the triple gene block and the binary movement block, two related transport gene modules, on the other hand. These transport systems use unrelated mechanisms of endomembrane reorganization and, accordingly, different virus transport pathways, which are based on either modification of the endoplasmic reticulum tubules, or formation of motile vesicles detached from the endoplasmic reticulum and targeted to endosomes. Based on the presented considerations, we emphasize that the mechanism of plant virus cell-to-cell transport is determined by the mode of virus interaction with cell endomembranes" I suggest: "Here, we discuss two distinct virus transport pathways based on either modification of the endoplasmic reticulum tubules or formation of motile vesicles detached from the endoplasmic reticulum and targeted to endosomes. The viruses with the movement proteins encoded by the triple gene block exemplify the first, and the potyviral system is the example of the second type. These transport systems use unrelated mechanisms of endomembrane reorganization. We emphasize that the mode of virus interaction with cell endomembranes determines the mechanism of plant virus cell-to-cell transport."

Changed as suggested.

Figure 01 is a very nice summary of the main idea of the review.

The figure 01 legend should not simply state "See text for details" but take advantage of the very nice visual tool to describe and summarize the review.

Red signs, question marks, and arrows should summarize the central idea of ​​the manuscript.

I suggest adding: "The two dissimilar pathways described are either modification of the endoplasmic reticulum tubules - TGB/BMB-specific (upper part) or formation of motile vesicles detached from the endoplasmic reticulum and targeted to endosomes - potyvirus-specific (bottom).

After this introduction, add a brief step-by-step description of each mechanism.

The figure legend is expanded as suggested.

Conclusions:

"Here, using the TGB/BMB-mediated virus transport and the potyvirus-specific transport as examples, we substantiate the idea that plant viruses can use dissimilar, unrelated mechanisms of modification of cell endomembranes for their movement in plants."

Show which virus groups these two described groups are representative of and why.

In the new version of Conclusions, to emphasize that possible transport mechanisms are not limited to the two pathway discussed in this review, we mention an additional virus group (carmoviruses), which likely have a distinct transport pathway unrelated to TGB/BMB and potyvirus pathways.

Reviewer 4 Report

Dear Authors,

I have read your manuscript review entitled " Dissimilar pathways of endomembrane system-dependent 2 plant virus movement" In general I find the work interesting, I appreciate the choosing uncovered and characterized in detail

But I differ with you in adopting all the reviews in only one Figure, while the goal is a detailed explanation on the intracellular transport of viral genomes to PD, we discuss two distinct TMV-unrelated plant virus transport systems, for which dissimilar mechanisms of reorganization of cell endomembranes for virus transport. This makes the reader bored without illustrations of a topic that aims to be detailed and clarified. Therefore, I ask the researchers to make several illustrative figures with short explanations in words to break the boredom and to make the topic interesting and complete.

- In conclusion, the authors add sentence about the applications of this work and the future vision.

Author Response

But I differ with you in adopting all the reviews in only one Figure, while the goal is a detailed explanation on the intracellular transport of viral genomes to PD, we discuss two distinct TMV-unrelated plant virus transport systems, for which dissimilar mechanisms of reorganization of cell endomembranes for virus transport. This makes the reader bored without illustrations of a topic that aims to be detailed and clarified. Therefore, I ask the researchers to make several illustrative figures with short explanations in words to break the boredom and to make the topic interesting and complete.

We agree with the Reviewer 2 that Fig. 1 provides a good summary of the main idea of the review and do not feel that additional illustrations might ‘make the topic interesting and complete’.

- In conclusion, the authors add sentence about the applications of this work and the future vision.

In the last phrase of Conclusions, the directions of future work are already outlined (‘detailed analyses of host membrane modifications observed for plant virus transport systems of other types…’). Besides, in the new version of Conclusions we emphasize the importance of studies of plant virus interaction with endomembranes for understanding the mechanisms of virus cell-to-cell transport.

Round 2

Reviewer 4 Report

 In my opinion, this version of the manuscript can be accepted for publication